# Monitoring Trends in Distribution and Seasonality of Medically Important Ticks in North America Using Online Crowdsourced Records from iNaturalist

**DOI:** 10.3390/insects13050404

**Published:** 2022-04-22

**Authors:** Benjamin Cull

**Affiliations:** Department of Entomology, University of Minnesota, St. Paul, MN 55108, USA; cull0122@umn.edu

**Keywords:** Ixodidae, distribution, expansion, tick-borne disease, blacklegged tick, lone star tick, Gulf Coast tick, American dog tick

## Abstract

**Simple Summary:**

An increasing number of cases of tick-borne diseases are being reported across North America and in new areas. This has been linked to the spread of ticks, primarily the blacklegged tick *Ixodes scapularis* and the lone star tick *Amblyomma americanum*, into new geographical regions. Tick surveillance systems have played an important role in monitoring the changing distributions of these ticks and have benefitted greatly from including data collected by members of the public through citizen or community science projects. Enlisting the help of community scientists is an economical way to collect large amounts of data over a wide geographical area, and participants can also benefit by receiving information relevant to their tick encounter, for example regarding tick-borne disease symptoms. This study examined tick observations from the online image-based biological recording platform iNaturalist to evaluate its use as an extra tool to collect information on expanding tick distributions. The distribution and seasonality of iNaturalist tick observations were found to accurately represent those of the studied species and identified potential new areas of tick expansion. Free-to-access iNaturalist data is a highly cost-effective method to support existing tick surveillance strategies to aid preparedness and response in emerging areas of tick establishment.

**Abstract:**

Recent increases in the incidence and geographic range of tick-borne diseases in North America are linked to the range expansion of medically important tick species, including *Ixodes scapularis*, *Amblyomma americanum*, and *Amblyomma maculatum.* Passive tick surveillance programs have been highly successful in collecting information on tick distribution, seasonality, host-biting activity, and pathogen infection prevalence. These have demonstrated the power of citizen or community science participation to collect country-wide, epidemiologically relevant data in a resource-efficient manner. This study examined tick observations from the online image-based biological recording platform iNaturalist to evaluate its use as an effective tool for monitoring the distributions of *A. americanum*, *A. maculatum*, *I. scapularis*, and *Dermacentor* in the United States and Canada. The distribution and seasonality of iNaturalist tick observations were found to accurately represent those of the studied species. County-level iNaturalist tick occurrence data showed good agreement with other data sources in documented areas of *I. scapularis* and *A. americanum* establishment, and highlighted numerous previously unreported counties with iNaturalist observations of these species. This study supports the use of iNaturalist data as a highly cost-effective passive tick surveillance method that can complement existing surveillance strategies to update tick distributions and identify new areas of tick establishment.

## 1. Introduction

In recent decades, there have been increases in the incidence and geographic ranges of tick-borne diseases in the United States (US) and Canada [1,2,3], linked to the geographic expansion of medically important tick species, primarily the blacklegged tick *Ixodes scapularis*, but also the lone star tick *Amblyomma americanum* and the Gulf Coast tick *Amblyomma maculatum*.

*Ixodes scapularis* is arguably the most important vector of tick-borne pathogens in North America, being the principal species responsible for transmitting *Borrelia burgdorferi*, the causative agent of Lyme disease, estimated to have caused almost 500,000 cases per year in the US during 2010–2018 [4]. This tick species also transmits *Anaplasma phagocytophilum* (human anaplasmosis), *Babesia microti* (babesiosis), *Borrelia mayonii* (Lyme disease), *Borrelia miyamotoi* (relapsing fever), *Ehrlichia muris eauclairensis* (human ehrlichiosis), and Powassan virus [5], which together cause thousands of human cases per year. In the US, the burden of *I. scapularis*-borne diseases falls primarily on the Northeastern and Upper Midwest regions, which have observed the greatest increases in *I. scapularis* range expansion in the last 20 years [6]. Establishment of *I. scapularis* in southern regions of Quebec, Ontario, Saskatchewan, New Brunswick, and Nova Scotia has resulted in increasing incidence of Lyme disease in Canada [3,7].

*Amblyomma americanum* is also a vector of multiple pathogens including *Ehrlichia chaffeensis* and *Ehrlichia ewingii* (human ehrlichiosis), *Francisella tularensis* (tularemia), Heartland virus, and Bourbon virus [1]. This tick species is also linked to the conditions Southern tick-associated rash illness (STARI) and alpha-gal syndrome. This tick’s core distribution is in the south-central and south-eastern areas of the US as well as along the eastern coast [8], and it appears to be expanding northward to reclaim its historical range [9]. This geographic expansion is thought to be associated with increases in the incidence of mild spotted fever group rickettsioses [10]; *Amblyomma americanum* is widely infected with *Rickettsia amblyommatis*, which has recently been demonstrated to be mildly pathogenic [11,12,13], and the tick is also a potential vector of both *R. rickettsii* [14,15] and *R. parkeri* [16,17]. The primary vector of *R. parkeri* is *A. maculatum*, which is expanding from its historical coastal distribution between Texas and Delaware, northwards through the central US and along the eastern coast [18]. In recent years, established populations of *A. maculatum* were reported in Connecticut [19], Illinois [20], and New York City [21], and a single questing specimen was collected in Indiana [22].

*Dermacentor andersoni* (Rocky Mountain wood tick) and *Dermacentor variabilis* (American dog tick) are also important vectors of *R. rickettsii* and *F. tularensis*, although their distributions have not changed dramatically compared to the species mentioned above. The range of *D. andersoni* includes most of the western US, and Alberta, British Columbia, and Saskatchewan in Canada [23,24]. Meanwhile, *D. variabilis* is widely distributed in the eastern US [25,26,27] and southeastern Canada, although there is some overlap of the two species in central areas.

Tick surveillance systems are essential for gaining the detailed knowledge of tick species, distribution, and seasonality that is required for health agencies to monitor the risk of transmission of tick-borne pathogens to humans and animals, and to provide information to the public on tick-borne disease risks and tick bite prevention measures.

Passive surveillance systems, in which tick specimens found on humans and animals are submitted to researchers by members of the public or veterinary and medical practices, have contributed significantly to knowledge of tick distribution, seasonal activity, host associations, and pathogen infection prevalence in North America, and can act as early warning systems for the introduction, establishment, and expansion of species of public health or veterinary concern [19,24,28,29,30,31,32,33,34,35,36,37,38,39,40,41,42,43,44,45,46,47,48,49,50,51]. Results from passive tick surveillance have been demonstrated to correlate well with the distribution of tick-borne diseases [7,29,45,52,53,54] and tick abundance from active surveillance [55].

The benefits of citizen or community science-driven vector surveillance approaches include: (a) data mainly describe human-vector encounters, making them epidemiologically relevant; (b) large amounts of data can be collected over a wide geographic area; (c) they are economical compared to active surveillance; (d) programs can be used to deliver reliable information about vectors of public health concern and how to reduce exposure; (e) relatively real-time access to data [56,57]. Meanwhile, drawbacks associated with these projects include: (a) lacking or incorrect information (particularly concerning geographical data); (b) biases because participants may not represent broader population; (c) submission of incorrect specimens.

Photograph-based passive surveillance systems such as the TickSpotters program [38,58], which involve crowdsourced submissions of tick images, have demonstrated high identification success of medically important tick species, and demonstrate potential to complement other tick surveillance approaches to monitor tick distributions and identify new foci. Since photographs are submitted directly through the program’s website, the cost to both participants and researchers is significantly lower compared to traditional passive surveillance, as no postage or specimen storage costs are involved, and the speed at which data can be received and reported are also improved.

An additional image-based approach that has shown potential to complement other tick surveillance strategies is the examination of tick images submitted to online wildlife databases such as iNaturalist [59]. The iNaturalist platform hosts a growing community of naturalists who submit primarily image-based wildlife observations via the iNaturalist smartphone app or website, where they can be identified by members of the community. The iNaturalist database currently contains over 92 million observations of over 344,300 species from across the globe. This valuable resource is being increasingly used in scientific research, and these studies demonstrate consistent results between community- and scientist-collected data [60,61,62,63,64,65,66,67,68,69,70]. A previous analysis found that iNaturalist data concerning important tick and mosquito vectors correlated well with known geographical distributions and seasonality of these species [59].

The purpose of this work was to employ tick data collected through iNaturalist to describe the distribution and seasonality of the medically important tick species discussed above, and further assess iNaturalist as a useful monitoring tool by comparing the data with other tick surveillance datasets from the US. iNaturalist tick data produced tick distribution maps consistent with the current known ranges of medically important tick species and was able to highlight potential new areas of expansion. Furthermore, the data accurately described regional differences in tick seasonality, and demonstrated good agreement with other tick surveillance datasets in areas of *I. scapularis* and *A. americanum* establishment. These findings demonstrate that iNaturalist is a useful additional tool that can be used to complement existing vector surveillance methods to describe distributions of key tick species and identify emerging foci at risk from tick-borne disease.

## 2. Materials and Methods

### 2.1. Study Region

The study region was chosen to include the known geographical range of *I. scapularis* [3,6] and all adjacent territories so that any evidence of expansion beyond its current range could be detected. This area also covers the known ranges of *A. americanum* [8,9], *A. maculatum* [18,19,20,21,22], and *D. variabilis* [25,26,27]. US state-level data were grouped into regions corresponding to the US Standard Federal Regions (Table 1). For this study, regions 1 and 2 were combined to create an equivalent-sized geographic area to the other regions. Canadian province-level data were grouped into three regions: Western (containing Saskatchewan and Manitoba), Central (Ontario and Quebec), and Atlantic (New Brunswick, Newfoundland and Labrador, Nova Scotia, and Prince Edward Island).

### 2.2. Data Analysis

iNaturalist observations typically include an image, with associated geographical location data and the time and date of the observation. Users may also add notes related to the observation, and annotations such as life stage and sex. Upon upload, an identification is suggested by iNaturalist based on matching the submitted image to identified images in the database. Once an observation obtains multiple agreeing species-level identifications from iNaturalist users it is labelled “research grade”, otherwise it is marked “needs id”. Observations lacking either a photo, location, or date are labelled as “casual”.

Data were downloaded directly from iNaturalist.org on a state-by-state basis via the ‘Identify’ page of the website. Searches were conducted for all observations of “Ticks (Order: Ixodida)” dated to the end of 2021. Observations labelled “casual” were excluded from the data. All images associated with observations were identified individually by a medical entomologist trained in the morphological identification of ticks, and any duplicate observations, images of too poor quality to identify to at least Ixodidae or Argasidae, and images of non-tick organisms were excluded from the dataset. Since the examination of spiracular plates is required for the separation of adult *Dermacentor* species [71] and the study region covers areas where *D. andersoni* and *D. variabilis* populations overlap in Saskatchewan, Montana, Wyoming, North Dakota, South Dakota, and Nebraska [23,24,25,26], a conservative approach was taken to identify *Dermacentor* to genus level only. Other ticks were identified to species level, if possible, by examination of standard morphological features.

Observations rarely included travel history, and therefore location data were accepted as accurate unless the observation included notes on recent travel contrasting with the recorded location. Tick observations were mapped in ArcGIS online (ESRI, Redlands, CA, USA). Comparative analyses of *I. scapularis* and *A. americanum* distributions utilized county-level data from US Centers for Disease Control and Prevention (CDC) studies [6,8,72] as a baseline, which was compared to iNaturalist tick observations as well as data from two recent US-wide passive tick surveillance programs from Northern Arizona University [31,45] and TickSpotters [38]. As this analysis only compared whether there was agreement between each surveillance dataset in reporting presence of the tick species in each county, it did not distinguish between established and reported populations [6,8,38]. County-level maps were created using mapchart.net.

## 3. Results

### 3.1. Overview of Tick Observations in Study Region

After removing casual observations, duplicates, non-tick organisms, and poor-quality images, which together accounted for 3% of all data, a total of 17,172 observations of ticks (Ixodida) were identified (Table 1). Although tick species were not separated by life stage for this study, it was noted that the vast majority of tick images exhibited adult stages. *Dermacentor* had the most observations (7594; 44.2%), followed by *Ixodes* (4523; 26.3%), *Amblyomma* (4484; 26.1%), *Haemaphysalis* (60; 0.3%), and *Rhipicephalus* (54; 0.3%). A number of images (436; 2.5%) were not of sufficient quality to identify beyond Ixodidae. Argasidae made up only 0.1% of all tick observations. There were 3372 observations identified as *A. americanum* and 879 as *A. maculatum*. Other *Amblyomma* species included *A. cajennense* s.l. (all records in southern Texas), and *A. dissimile* and *A. tuberculatum* (both species in Florida). Observations of *Ixodes* included 4362 *I. scapularis*, as well as *I. brunneus*, *I. cookei*, *I. marxi*, and *I. uriae*. *Haemaphysalis* observations included *H. longicornis* (identified in New York, New Jersey, Maryland, Pennsylvania, and Virginia), *H. chordeilis*, and *H. leporipalustris*. Species-level identifications of *Rhipicephalus* were all *R. sanguineus* s.l. Although they were not identified to species, the majority of *Dermacentor* records likely represent *D. variabilis* based on the study region, with those observations in the westernmost areas of the region likely *D. andersoni*. Furthermore, a number of *Dermacentor* observations could be identified as likely *D. albipictus* based on their association with moose (*Alces alces*).

The earliest dated tick observation was from 1997, but 96% observations were from the last five years and 39% were from 2021 alone (Figure 1), indicating that iNaturalist data primarily constitute recent tick sightings. Observations have increased each year, which is more likely an effect of the increasing popularity of iNaturalist rather than significant increases in tick abundance or human-tick exposure.

### 3.2. Distribution of iNaturalist Observations of Target Tick Species

The distribution of iNaturalist observations of the tick species examined are consistent with their known distributions in North America (Figure 2). Observations of *A. americanum* primarily occurred in the south, southeast, and eastern coast of the US, with scattered observations in more northerly states of the upper Midwest and Northeast and the lower areas of Ontario (Figure 2A), reflecting recent tick surveillance data [8,38,49]. Similarly, *A. maculatum* observations were typically within the known range in US states along the southern and southeastern coast, and south-central areas [18], but with evidence of northward expansion into Nebraska, Missouri, Illinois, Indiana, Ohio, New Jersey, and New York, consistent with recent reports [19,20,21,22]. Scattered observations at the northern edge were also recorded in Iowa, Michigan, and Ontario (Figure 2B). Observations of *Dermacentor* spp. were broadly distributed throughout the US portion of the study region, and in Saskatchewan, Nova Scotia, New Brunswick, and the more southerly areas of Manitoba, Ontario, and Quebec in Canada (Figure 2C). Again, observations of *I. scapularis* were found throughout its known established range with scattered observations at the edges (Figure 2D), suggestive of expansion into new areas, as reported in the literature [3,6,38,45,47,48,54,73,74].

To determine the relative risk of human exposure to different tick species in different regions, the proportion of each species reported by iNaturalist users was examined (Figure 3A). Consistent with the distribution of *A. americanum*, the proportion of observations of this tick were greatest in the central, southern, and southeastern regions (US Regions 4, 6, and 7). Similarly, the proportion of *A. maculatum* observations was highest in US Regions 6 and 4. The greatest proportions of *I. scapularis* observations occurred in US Regions 1 and 2, and in central and western Canada. Interestingly, despite *I. scapularis* being prevalent in the upper Midwest (US Region 5), a much higher proportion of observations from this region were of *Dermacentor* spp. The proportions of *A. americanum*, *Dermacentor* spp., and *I. scapularis* observations were approximately equal in US Region 3. As would be expected based on the overall numbers of tick observations (Table 1), *Dermacentor* spp. made up a relatively large proportion of observations across all regions, particularly those where *I. scapularis* is yet to establish widely and the *Amblyomma* species are so far absent (US Region 8, Western, and Atlantic Canada). The proportions of different tick species remained relatively stable between years 2017–2021 in each region, suggesting no obvious increases in observations of any tick species relative to the others.

### 3.3. Seasonality of Tick Observations

Observation data from iNaturalist were examined to determine the seasonality of the different tick species in each region (Figure 3B). In more northerly US states, the seasonal activity of *A. americanum* was observed from March to August, peaking in May/June, whereas in southern regions, the activity began earlier in the year (February) and peaked April/May, with an extended duration of activity observed into September and October. The highest observations of *A. maculatum* were typically reported June to August, but in US Region 6, observations of this species were made throughout the year, peaking in July to September. Observations of *Dermacentor* spp. demonstrated a consistent pattern across the study region, with activity between March and August, peaking May/June. This pattern was different in US Region 6, where activity was extended February to October, with the maximum observations in April. Seasonal variation in *I. scapularis* observations indicated bimodal activity, with an early peak in spring/summer and a late peak in October/November. In southern regions, *I. scapularis* observations were made throughout the winter months at the start and end of each year, with reduced activity in the summer.

### 3.4. Comparison of County-Level Distributions of I. Scapularis and A. Americanum

To aid the assessment of tick-borne disease risk to the US population, the distribution of tick species in the continental US is often mapped at a county-level [6,8,26]. Therefore, to evaluate the effectiveness of iNaturalist data as a useful tool for tick surveillance, county-level observations for *I. scapularis* and *A. americanum* were compared to those from CDC data compiled from literature reviews, national collections, and vector surveillance programs [6,8,72], as well as two recent US-wide passive surveillance programs: a citizen science tick collection project run by Northern Arizona University in which tick specimens were submitted by the public for tick testing [31,45], and the photo-based TickSpotters project from the University of Rhode Island Tick Encounter Resource Center [38]. There was high agreement among the different datasets in the distribution of counties with records of *I. scapularis* in the upper midwest and northeast of the country (Figure 4A), whereas areas of high agreement in the southeastern US were more scattered.

Each surveillance dataset had counties with *I. scapularis* reported that were not present in other datasets (Table 2 and Table 3). County-level records from iNaturalist added an additional 49 counties where *I. scapularis* was not reported in other datasets. These were located in areas adjacent to those with previous reports of *I. scapularis*, and were primarily located in Texas (nine counties), Tennessee (seven counties), Alabama (five counties), Missouri (five counties), and at the northwestern edge of *I. scapularis* range in North and South Dakota and western Minnesota (Figure 4A). iNaturalist observations of *I. scapularis* occurred in 37% counties reported in CDC data, 56% counties reported by TickSpotters, and 62% counties reported in Northern Arizona University surveillance data (Table 3). Data from this latter project performed similarly to iNaturalist when compared against other datasets, whereas TickSpotters demonstrated better overall coverage. The three passive surveillance schemes failed to record *I. scapularis* in the majority of counties where this tick is established or reported according to the CDC dataset, with even the best performing passive scheme overlapping with less than half of CDC-reported counties. However, all three passive schemes added counties that were not represented in CDC data (Table 3). To test the accuracy of iNaturalist in reporting counties with *I. scapularis*, a confusion matrix was constructed of iNaturalist data against agreeing data from the other US tick datasets. This identified a total of 1819 counties, where the three existing datasets agreed: 502 with and 1317 without *I. scapularis* records. iNaturalist agreed with 380/502 (76%) counties where *I. scapularis* is reported present and with 1268/1317 (96%) counties where the tick is reported absent, giving an accuracy of 0.91 and a *kappa* statistic of 0.75. As mentioned above, iNaturalist further identified 49 counties with previously unreported *I. scapularis.*

For *A. americanum* there was high agreement among datasets in the south-central, southeastern, and eastern regions of the US (Figure 4B). There were 61 counties where this tick species was reported by iNaturalist users where it was not present in other datasets examined (Table 2). As with *I. scapularis*, these were near to areas where *A. americanum* has already been reported by other surveillance programs (Figure 4B), with highest numbers in Illinois (12 counties), Kentucky (eight counties), and Oklahoma (seven counties). Further previously unreported counties were distributed across the edges of the *A. americanum* range in Kansas, Nebraska, Iowa, Minnesota, Wisconsin, and Michigan (Figure 4B).

Counties with iNaturalist observations of *A. americanum* had a similar rate of overlap with existing tick datasets as for *I. scapularis*, agreeing with 42% counties reported in CDC data, 53% counties reported by TickSpotters and 65% counties reported by Northern Arizona University (Table 4). Data from the three passive surveillance projects contributed a large amount of new county occurrence data for *A. americanum*, with 24–33% of counties from these datasets not represented in CDC baseline data (Table 4). Each dataset contributed unique occurrence records in multiple counties (Table 2).

## 4. Discussion

This study evaluated freely available data from the iNaturalist online community on observations of medically important tick species across a large region of North America and found that results regarding the distribution and seasonality of ticks were consistent with the current knowledge on the species examined. Additionally, a small proportion of previously unreported county occurrence records of both *A. americanum* and *I. scapularis* were documented, suggesting that this easily accessible data source could be a useful passive surveillance method to be employed alongside other vector surveillance strategies to corroborate distribution data and identify areas of emerging tick establishment and subsequent disease risk.

A discussion of the use of iNaturalist data in the context of vector surveillance has been provided in a previous analysis [59], and this method shares many of the benefits and drawbacks of other passive tick surveillance programs [56,57]. On the plus side, these include resource-efficient collection of large datasets covering a wide geographical area, epidemiologically relevant data describing human-tick encounters, and real-time access to tick observations. The downsides include missing information accompanying records and the potential for inaccurate geographical location data due to users reporting a location distant from the site of tick acquisition. The use of iNaturalist also has various advantages and disadvantages compared to other passive tick surveillance methods. A major advantage is the open access nature of the dataset, which makes this an extremely cost-effective method for examining tick occurrence data. In comparison, even the most resource-efficient existing passive tick surveillance system requires operation costs to administer and publicize the program, communicate results, maintain a public-facing website, store specimens, etc. As a general wildlife observation platform, iNaturalist also includes a vast amount of data of other species that may be relevant to tick research, for example animals that are important tick hosts and/or reservoir hosts for tick-borne pathogens (Figure 5). One study has utilized iNaturalist observations to record tick infestations on rare Southern Alligator Lizards (*Elgaria multicarinata*) [61]. This aspect of iNaturalist also means that users need no pre-existing tick awareness to report a tick observation, as an image can be uploaded without knowledge of its identity. On the other hand, since it is not a dedicated tick recording system, the accompanying data that are usually collected as part of passive tick surveillance, such as tick host, are not routinely provided in iNaturalist records. One major benefit of most dedicated passive tick surveillance programs is the provision of public health information relating to tick bite prevention and tick-borne diseases, which is outside the remit of a platform such as iNaturalist.

A downside associated with photo-based passive surveillance systems is that tick specimens are not available for pathogen testing or confirmation of species identification, although the use of images increases the speed of data acquisition and reduces costs associated with postage and storage. The TickSpotters program highlights that identification success of images of key species of medically important ticks can be high, reporting 98–99% accuracy for *I. scapularis*, *D. variabilis*, and *A. americanum* [58], although this program gives detailed guidelines for submitters to ensure clear images are provided with key identification features visible [38]. Images submitted to iNaturalist are of variable quality, ranging from blurry/out of focus photos that cannot be identified to high quality images taken under a microscope, and this range of image quality is likely to lead to an overall lower identification rate. Despite this, a large dataset of ticks identifiable to species level is available, as can be observed from this study, and only a relatively low percentage of observations (3%) needed to be excluded. The iNaturalist data examined for this study contained a total of 4362 identified *I. scapularis* images, compared to 6429 *I. scapularis* collected during 2016–2019 by the Northern Arizona University project [45] and 9532 collected through TickSpotters 2014–2019 [38]. The iNaturalist data included 3372 identified *A. americanum* observations, whereas TickSpotters received 5746 records and Northern Arizona University reported 2078 over two years [31]. Whilst iNaturalist datasets covering similar time periods are comparably smaller, the rapid year-on-year increases in tick observations by the community suggest that this dataset will continue to grow. Although slightly quicker than the morphological identification of tick samples due to the time associated with specimen handling, the identification of ticks via photographs is still a time-consuming process, and advances in automated tick identification [75] could significantly improve speed in the future. To gain a rapid overview of tick data, it is also possible to rely on the iNaturalist community identifications, which are generally good, although less common species may be misidentified; for example, during this study it was noticed that *A. maculatum* observations were sometimes misidentified as *Dermacentor*, particularly in areas where the Gulf Coast tick is less common.

The distribution of tick species derived from iNaturalist data demonstrated a good approximation with their known ranges, with most evidence of expansion reflected in current literature. For example, the occurrence of records of *A. maculatum* in Illinois and New York corresponds with recent evidence of established populations in these states [20,21]. It is unclear whether observations of *A. maculatum* in Michigan and Ontario represent travel-associated ticks or a further northward expansion of this species. Similar to recent passive tick surveillance projects [38,45], this study identified a number of US counties with previously unreported *A. americanum* and *I. scapularis* records from iNaturalist observations. These were primarily in counties adjacent to areas with established or reported tick presence, suggesting that these likely represent areas of tick expansion such as in eastern South Dakota, where *I. scapularis* has been confirmed by active field surveillance in recent years [73,74]. However, since geographical data associated with iNaturalist observations were not verified, it is not possible to determine whether these simply resulted from travel to neighboring counties. There was good county-level agreement for *I. scapularis* in the northeast and upper midwest regions of the US, and for *A. americanum* in the southeastern and eastern US, corresponding with the currently understood high risk zones for encountering these tick species. Notably, states with most novel county reports of *I. scapularis* (Texas) and *A. americanum* (Illinois and Kentucky) identified in iNaturalist data were also among states found to have the most new county reports by TickSpotters [38], suggesting these are areas where the species are expanding or else historically under-sampled. Whilst *Dermacentor* were not identified to species, since both *D. andersoni* and *D. variabilis* act as vectors for *R. rickettsii* and *F. tularensis*, and the number of tentatively identified *D. albipictis* was small, the iNaturalist data-derived map may represent a good risk map of exposure to medically important *Dermacentor* species. Citizen science-derived data represent human observations, and therefore they can be biased towards areas of high population density [48,76]. Similarly, in a previous analysis of iNaturalist tick observations in the state of Minnesota, the majority of observations were located in the highly populated Greater Minneapolis–St. Paul Metropolitan area [59]. In the current study, there was a positive correlation of the number of tick observations per state/province with human population size (R^2^ = 0.493), but not with population density (R^2^ = 0.005), supporting a role of human population size in determining the success of detecting tick species through passive surveillance.

Most of the iNaturalist observations were adult specimens, presumably because these are easier to detect and photograph, and this passive surveillance method is therefore less effective at collecting information on immature tick stages. As a result, the seasonal observations of ticks gained from iNaturalist data describe the documented phenology of the adult stages of tick species in North America [1,22,34,40,41,49,50,51,77,78]. This will have implications for the suitability of tick occurrence data from iNaturalist in informing the risk from diseases that are epidemiologically linked to nymphal stages of ticks. Further identifying iNaturalist observations to life stage would allow the separation and seasonal recording of adult and immature ticks.

## 5. Conclusions

In conclusions, this study supports the use of openly accessible iNaturalist data as a highly cost-effective method of passive tick surveillance that can complement existing surveillance strategies to update tick distributions and identify new areas at risk of tick establishment that could be further targeted with active surveys and tick awareness information. Whilst the utility of iNaturalist data as a standalone vector surveillance tool is perhaps limited, it is valuable for contributing additional data to enhance and validate other surveillance and biodiversity datasets (such as Global Biodiversity Information Facility (GBIF)). Furthermore, this study highlights the value of combining various surveillance strategies to improve knowledge of established and emerging areas of tick occurrence. This integrated surveillance could take a similar approach to the algorithm suggested by Nelder et al. [49] to monitor *A. americanum* risk areas. Vector control and public health agencies could, with little effort and at little additional expense, incorporate a weekly or monthly check of iNaturalist tick records in their jurisdiction, which would provide additional data to support existing passive and active surveillance activities.

## Figures and Tables

**Figure 1 insects-13-00404-f001:**
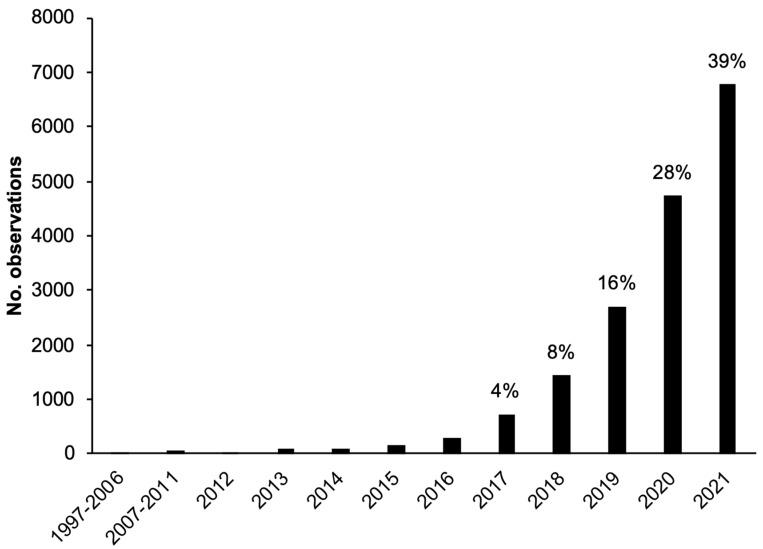
Number of iNaturalist tick observations in the study area separated by year. The percentage of total tick observations for each year is labelled above each bar for 2017–2021.

**Figure 2 insects-13-00404-f002:**
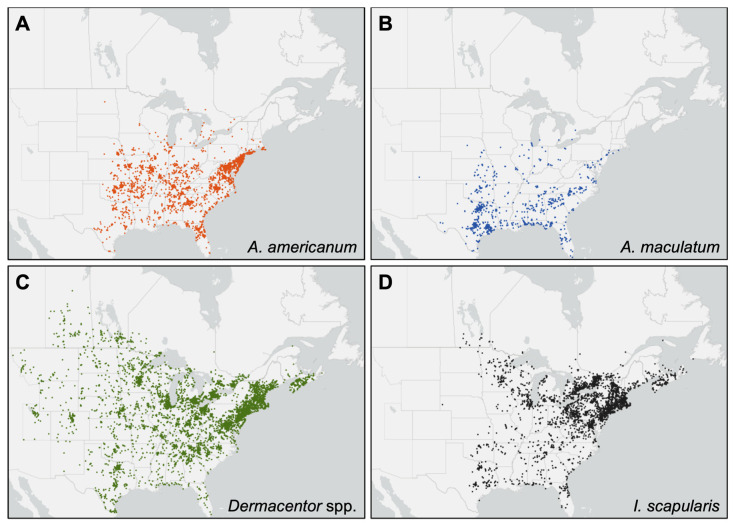
Distribution of iNaturalist observations of target tick species: (**A**) *Amblyomma americanum*; (**B**) *Amblyomma maculatum*; (**C**) *Dermacentor* spp.; (**D**) *Ixodes scapularis.* Data used to create these distribution maps can be found in Appendix A.

**Figure 3 insects-13-00404-f003:**
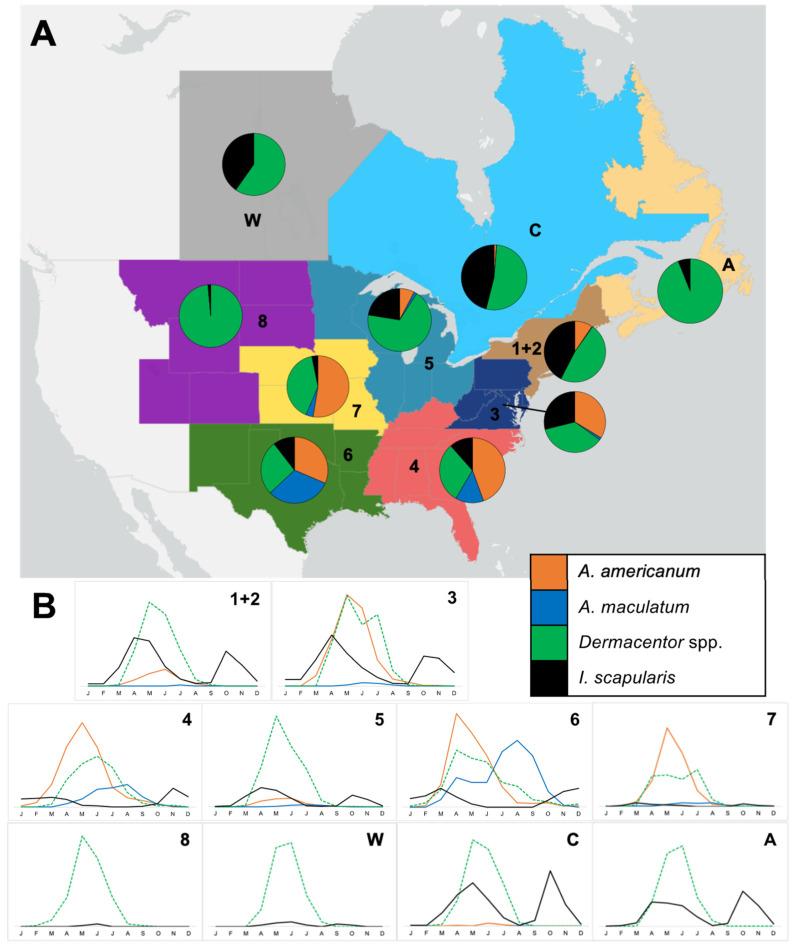
(**A**) Proportion and (**B**) seasonality of iNaturalist observations of target tick species by region. Numbers and letters refer to the Regions defined in Table 1. The *x* axis represents no. observations of each species.

**Figure 4 insects-13-00404-f004:**
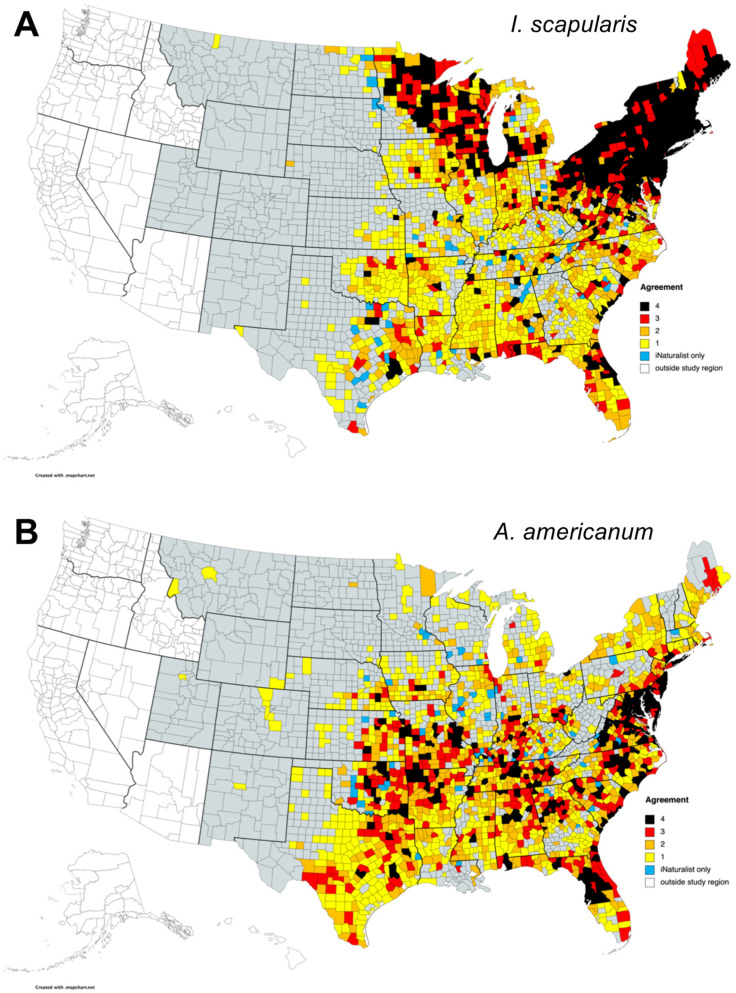
Comparison of recorded United States county-level distribution of (**A**) *Ixodes scapularis* and (**B**) *Amblyomma americanum* between iNaturalist data and recent sources of US-wide tick surveillance data Refs. [6,8,31,38,45,72]. The number of surveillance methods reporting tick presence in each county is indicated by the color. [72] includes data from the ArboNET Tick Module. Maps created with mapchart.net. Raw data are available in Appendix A.

**Figure 5 insects-13-00404-f005:**
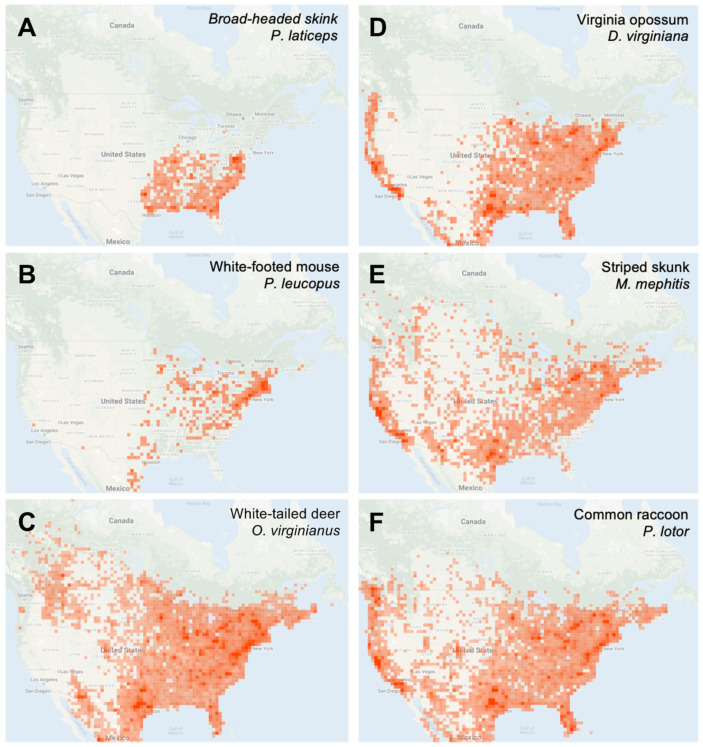
Distribution of iNaturalist research grade observations of various small, medium, and large animal hosts of ticks: (**A**) *Plestiodon laticeps*; (**B**) *Peromyscus leucopus*; (**C**) *Odocoileus virginianus*; (**D**) *Didelphis virginiana*; (**E**) *Mephitis mephitis*; (**F**) *Procyon lotor*. Map data Copyright 2022 Google, INEGI.

**Table 1 insects-13-00404-t001:** Summary of iNaturalist tick observations by region, state/province, Family, and Genus.

Region	State/Province	Ixodidae *						Argasidae	Total
		*Amb*	*Der*	*Hae*	*Ixo*	*Rhi*	Unidentified		
US Region 1	Connecticut	4	133	0	111	0	1	0	249
	Maine	0	167	0	48	0	7	0	222
	Massachusetts	23	458	0	390	0	16	0	887
	New Hampshire	0	240	0	83	0	1	0	324
	Rhode Island	1	23	0	23	0	1	0	48
	Vermont	0	193	2	359	0	9	0	563
US Region 2	New Jersey	255	361	9	142	0	27	0	794
	New York	123	348	22	589	0	29	0	1111
US Region 3	Delaware	63	33	0	5	0	4	0	105
	District of Columbia	21	7	1	7	0	4	0	40
	Maryland	250	183	6	110	0	26	1	576
	Pennsylvania	33	297	9	424	0	15	0	778
	Virginia	501	290	5	118	0	29	0	943
	West Virginia	6	51	0	42	0	2	0	101
US Region 4	Alabama	176	66	0	29	0	14	0	285
	Florida	329	40	0	60	5	12	0	446
	Georgia	179	64	0	28	0	7	0	278
	Kentucky	109	90	0	27	0	4	1	231
	Mississippi	79	27	0	16	0	5	0	127
	North Carolina	217	181	1	33	0	22	0	454
	South Carolina	89	38	0	18	0	6	0	151
	Tennessee	171	153	0	42	0	8	0	374
US Region 5	Illinois	98	380	0	81	0	13	0	572
	Indiana	69	114	0	29	0	4	0	216
	Michigan	6	277	0	95	0	9	0	387
	Minnesota	2	307	0	100	0	3	2	414
	Ohio	58	475	2	177	0	20	0	732
	Wisconsin	5	171	0	100	0	2	0	278
US Region 6	Arkansas	125	25	0	10	0	9	0	169
	Louisiana	61	16	0	14	0	1	0	92
	New Mexico	0	6	0	0	2	3	5	16
	Oklahoma	230	76	0	24	0	5	1	336
	Texas	571	258	0	107	46	27	5	1014
US Region 7	Iowa	16	70	0	18	0	3	0	107
	Kansas	72	50	0	3	0	5	3	133
	Missouri	475	110	0	20	0	20	3	628
	Nebraska	43	197	0	1	0	4	0	245
US Region 8	Colorado	1	63	0	2	0	2	0	68
	Montana	0	45	0	0	0	0	0	45
	North Dakota	1	53	0	3	0	1	0	58
	South Dakota	0	52	0	1	0	0	0	53
	Utah	1	44	0	1	1	5	0	52
	Wyoming	0	24	0	0	0	2	0	26
Atlantic	New Brunswick	0	12	1	31	0	1	0	45
Canada	Prince Edward Island	0	0	0	5	0	0	0	5
	Nova Scotia	0	289	1	178	0	19	0	487
	Newfoundland and Labrador	0	0	0	4	0	2	0	6
Central	Ontario	21	829	0	739	0	25	0	1614
Canada	Quebec	0	20	1	60	0	0	0	81
Western	Manitoba	0	95	0	16	0	1	0	112
Canada	Saskatchewan	0	93	0	0	0	1	0	94
	TOTALS	4484	7594	60	4523	54	436	21	17,172

* *Amb*: *Amblyomma*; *Der*: *Dermacentor*; *Hae*: *Haemaphysalis*; *Ixo*: *Ixodes*; *Rhi*: *Rhipicephalus*. Full data can be found in Appendix A.

**Table 2 insects-13-00404-t002:** Comparison of county-level occurrence data of *I. scapularis* and *A. americanum* in US states included in the study region from various US tick surveillance data sources. Data are available in Appendix A.

Surveillance Data Source	Time Period	No. Counties with *I. scapularis*	No. Unique County Records	No. Counties with *A. americanum*	No. Unique County Records
Eisen et al., 2016 [6]; CDC, 2021 [72]	1996–2020	1570	605	-	-
Springer et al. 2014 [8]	1898–2012	-	-	1292	454
Northern Arizona University [31,45]	2016–2019 (*Is*) * 2016–2017 (*Aa*)	688	33	438	34
TickSpotters [38]	2014–2019	896	73	1024	181
iNaturalist [this study]	1997–2021	679	49	727	61

* Data available 2016–2019 for *I. scapularis* [45], and 2016–2017 for *A. americanum* [31].

**Table 3 insects-13-00404-t003:** Overlap of counties with reports of *I. scapularis* between various US tick surveillance data sources. County-by-county comparison data are available in Appendix A.

	CDC	TickSpotters	Northern Arizona University	iNaturalist
iNaturalist	587/1570 (37%)	504/896 (56%)	428/688 (62%)	-
Northern Arizona University	616/1570 (39%)	536/896 (60%)	-	428/679 (63%)
TickSpotters	762/1570 (49%)	-	536/688 (78%)	504/679 (74%)
CDC	-	762/896 (85%)	616/688 (90%)	587/679 (86%)

**Table 4 insects-13-00404-t004:** Overlap of counties with reports of *A. americanum* between various US tick surveillance data sources. County-by-county comparison data are available in Appendix A.

	CDC	TickSpotters	Northern Arizona University	iNaturalist
iNaturalist	537/1292 (42%)	540/1024 (53%)	285/438 (65%)	-
Northern Arizona University	332/1292 (26%)	352/1024 (34%)	-	285/727 (39%)
TickSpotters	691/1292 (53%)	-	352/438 (80%)	540/727 (74%)
CDC	-	691/1024 (67%)	332/438 (76%)	537/727 (74%)

## Data Availability

The data presented in this study are available in Appendix A. The original data underlying the presented results are available from iNaturalist.org, accessed on 20 March 2022.

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
