# Peer review of "Monitoring Trends in Distribution and Seasonality of Medically Important Ticks in North America Using Online Crowdsourced Records from iNaturalist"

_insects, 2022, doi:10.3390/insects13050404_

Round 1
Reviewer 1 Report
This is an interesting topic and clearly INaturalist is a good vehicle to test whether CitSci data are a useful and reliable addition to surveillance and monitoring. That data provided here at seem to be quite a convincing demonstration that iNaturalist data is reliable, and matches the other available datasets.
BUT there are some potentially critical bits of info left out - most importantly how many data were discarded during the validation process. if thats a high proportion than the value of the dataset is more questionable
Also, though there are a small (and un-reported) proportion of records unique to iN , the tables suggest that iN has fewer records than the other datasets which which it has been compared. It would be useful to know where iN does not record data where the other datasets do. In that context the comparisons discussed are all visual with no actual analyses. Perhaps a simple confusion matrix would help here.
My general conclusion is that the iN data is OK bit doesnt really provide anything that the others dont. Obviously its value includes advocacy an good way to show citizens what is going on and so on but perhaps its value in academic terms is when added to the other big datasets as validation and enhancement. Related to that, its a shame that the cleaned point location data arent provided - (or are they already in the public domain??)
I have added quite a few comments to the attached file which I hope are helpful.

Author Response
I appreciate the reviewer's careful consideration of this work, and they have raised many useful points that have helped to improve the paper. Thank you.
Please find responses to comments below:
line 48 - added "tick-borne" to the sentence
line 97 - made the recommended changes to this area on benefits of citizen science.
species names have been italicised throughout.
You make an excellent point about including the rejected data. I regret that I did not keep track of this during the analysis! But going back to the raw data I found that about 3% of total observations were excluded during analysis, and I have included this information in the first sentence of the Results. This is consistent with my previous publication that found 1.3 - 7.8 % iNaturalist records were excluded during analysis of smaller tick datasets (state level).
line 263 - deleted "further" to improve accuracy of the statement.
I have included a confusion matrix analysis and some additional tables comparing the overlap between the different datasets, and these reveal some interesting information about the agreement between different datasets and highlight the value of combining data from various sources. A full county by county comparison of all data sets can be found in Table S3 (previously S2).
Line 301 - corrected the sentence to say a "small proportion of previously unreported counties".
I agree with the reviewer's general conclusions and have added some sentences to the final paragraph to convey these points.
Following the reviewer's recommendation, I have provided a table (Table S2) of the cleaned point data used to create the maps in Figure 2.
Reviewer 2 Report
Peer review report on the manuscript "Monitoring trends in distribution and seasonality of medically important ticks in North America using online crowdsourced records from iNaturalist", (Manuscript ID: insects-1690676)
Recommendation: Accept
Comments to Authors:
This manuscript evaluates the photograph-based observations of ticks gained from the crowdsourced passive recording platform iNaturalist as a useful additional monitoring tool. The results of the tick images submissions were compared with the data from other tick surveillance datasets and produced new tick distribution maps consistent with the currently known ranges of medically important tick species.
The findings demonstrated that the passive tick surveillance data from the iNaturalist is a highly cost-effective method that showed good agreement with other data sources in documented areas of certain tick species.
Tick occurrence data from iNaturalist could be a valuable tool to complement existing vector surveillance methods to describe the distribution and seasonality of ticks, describe potential new areas of expansion, and identify emerging foci at risk from tick-borne diseases.
The paper is well written with a carefully organized text, the results are sufficiently presented and analyzed, and the discussion is valuable for a better understanding of the results.
For that reason, the paper makes a substantial contribution to the literature and is therefore recommended for publication in Insects after minor revision taking into account the following specific comment.
Specific comment
The scientific names of the tick’s species or other animals should always be italicized. In addition, the genus name shall always be capitalized, and the specific epithet that follows the genus name shall not be capitalized.
Please correct the scientific names of the species on lines 134 to137, 180 to 194, 204, and all the scientific names in the References section.
Author Response
I would like to thank the reviewer for their positive feedback on the manuscript.
Thank you for spotting these - I have corrected the errors associated with species names throughout the manuscript.